# Cellular Delivery of Functional AntimiR Conjugated to Bio-Produced Gold Nanoparticles

**DOI:** 10.3390/ncrna11050066

**Published:** 2025-09-11

**Authors:** Parastoo Pourali, Veronika Benson

**Affiliations:** 1Department of Chemistry, University of Wyoming, 1000 E. University Ave., Laramie, WY 82071, USA; ppourali@uwyo.edu; 2Laboratory of Immunotherapy, Institute of Microbiology of the Czech Academy of Sciences, 14220 Prague, Czech Republic; 3Faculty of Health Studies, Technical University of Liberec, 46001 Liberec, Czech Republic

**Keywords:** biologically produced gold nanoparticles, clathrin-mediated endocytosis, caveolin-mediated endocytosis, macropinocytosis, chemical blockers

## Abstract

**Background/Objectives**: Bio-produced gold nanoparticles (AuNPs) are effective carriers of short RNAs into specialized mammalian cells. Their potential application is still limited by scarce knowledge on their uptake and intracellular fate. Gold nanoparticles that are not biologically produced (NB-AuNPs) enter specialized cells primarily via clathrin-dependent endocytosis. Unlike the NB-AuNPs, the bio AuNPs possess natural surface coatings that significantly alter the AuNPs properties. Our research aimed to reveal the cellular uptake of the AuNPs with respect to delivering a functional RNA cargo. **Methods**: The AuNPs were conjugated with short inhibitory RNA specific to miR 135b. Mammary cancer cells 4T1 were pretreated with inhibitors of caveolin- and clathrin-mediated endocytosis and macropinocytosis. AuNPs’ uptake, fate, and miR 135b knock-down were assessed with TEM and qPCR. **Results**: The AuNPs-antimiR 135b conjugates entered 4T1 cells via all the tested pathways and could be seen inside the cells in early and late endosomes as well as cytoplasm. In contrast to the clathrin-dependent pathway, the caveolae-mediated endocytosis and the macropinocytosis of the AuNPs resulted in the effective targeting and reduction of the miR 135b. **Conclusions**: The bio-produced AuNPs can effectively enter mammalian cells simultaneously by different endocytic pathways but the delivery of functional cargo is not achieved via the clathrin-dependent endocytosis.

## 1. Introduction

Recent strategies in cancer chemotherapy use a so-called multi-target approach, in which cancer cells are targeted while non-cancerous cells survive [1]. This strategy also helps to reduce potential cancer resistance [2]. The multiple-target therapy involves binding therapeutic molecules with a carrier such as nanoparticles (NPs), which increases their circulation time in plasma and enhances their delivery to cancer cells. Biologically produced gold nanoparticles (AuNPs) show great potential as carriers for the multi-target tumor therapy [3]. The main advantage of AuNPs over non-biologically produced gold NPs (NB-AuNPs) is their lower toxicity and ability to easily conjugate with desired molecules due to the presence of capping agents on the AuNPs’ surface [3]. The molecules of the AuNPs’ capping agents, such as peptides or amino acids [4], can be attached to cargoes like proteins or nucleic acids through electrostatic interaction without the need for additional chemicals or procedures. The potential use of the AuNPs as drug carriers is also supported by their detection in various compartments of cancer cells following in vivo applications [3].

In contrast to NB-AuNPs, the intracellular fate, cellular internalization, and trafficking of the AuNPs is unclear. The cellular uptake and passage through the cytoplasmic membrane of the AuNPs are influenced by the NPs’ size, charge, shape, and many other factors [5]. The understanding of the AuNPs’ internalization route is important not only with respect to their safety, but also with respect to their integrity and ability to deliver intact cargoes to the cells. We need to know whether the AuNPs reach specific cells through any means of internalization and whether the conjugated cargo is still active enough to act on its target. We showed that changing the environmental pH to acidic levels, such as pH 2 or 4, caused AuNPs to agglomerate [6]. In the current study, we did not attempt to change the surface charge of the AuNPs; instead, we aimed to investigate the mechanism of their cell membrane internalization while their conjugated cargo remains active. The AuNPs allowed us to directly conjugate desired molecules to their surface without altering their charge, thereby enabling passage through cell barriers. Therefore, rather than altering the surface charge of the AuNPs, it is preferable to conjugate them with molecules for targeting specific cells or the cell nucleus using various cell-penetrating peptides (CPPs) [7].

So far, the main mechanism how the NPs enter mammalian cells is endocytosis, which is followed by NPs budding from endocytic vesicles and their aiming for intracellular location. Endocytosis is classified into two main types: (1) pinocytosis, which occurs in many types of cells and consists of caveolin-mediated, clathrin-mediated, and clathrin-/caveolae-independent endocytosis and macropinocytosis; and (2) phagocytosis, which mainly occurs in phagocyte cells [5,8,9]. The other possible mechanism for AuNP entry is passive penetration, such as diffusion, which involves the interaction of the AuNPs with the lipid bilayer of the cell membrane followed by the direct penetration of the AuNPs without compromising the cell’s integrity [5,10].

A particular challenge is represented by the in vivo application of any NPs. Like many other carriers entering body, the AuNPs are usually surrounded by certain serum proteins, generally referred to as the “protein corona”. The protein corona can change properties of the AuNPs. Once AuNPs reach and penetrate a specific organ, they must interact with the cellular microenvironment, which possesses different proteins or pH milieu compared to the bloodstream. All of these factors can potentially have either positive or negative impact on the cellular uptake and fate of the AuNPs [5]. Previously, we incubated the AuNPs with different concentrations of plasma and demonstrated that fibrinogen (Fg) was the main protein corona present in all concentrations used [4]. Although coating the AuNPs’ surface with Fg to determine their cellular fate is an interesting topic, in this study, we have used the AuNPs alone to achieve the fundamental results.

Our research aimed to understand the cellular uptake, intracellular transport, and fate of the AuNPs conjugated to specific cargo with respect to effective cargo delivery to cells. This process may differ from that of NB-AuNPs due to their specific surface coating, potentially impacting their medical applications. To achieve this goal, the AuNPs were bio-produced using *Fusarium oxysporum* and used as carriers in the murine 4T1 cell line, since these cells express high amounts of miR 135b. Also, this model cell line was used in many of our previous studies, thus allowing more complex understanding [3,11,12]. The AuNPs were conjugated with both antimiR 135b and transferrin (Tf). AntimiR 135b was used as an effector cargo here that needed to be delivered intact to the cells, find its target, and reduce the level of miR 135b. We showed previously that antimiR 135b can conjugate to the AuNPs directly and transfect the cells [12]. Tf was employed to actively induce receptor-mediated endocytosis through uptake by the Tf receptor (TfR), which is part of the clathrin-mediated endocytosis pathway [13]. Different chemicals were used to block various pathways of AuNP-antimiR 135b endocytosis, including the caveolin-mediated pathway, clathrin mediated pathway, and macropinocytosis, in both the Tf-conjugated and non-conjugated forms. To characterize the AuNPs-antimiR 135b (with or without the Tf) regarding their internalization efficacy and their ability to deliver the intact cargo to the cells, we employed two key methods: (1) TEM showing the distribution and amounts of AuNPs inside the cells and (2) qPCR of miR 135b demonstrating the effective internalization and delivery of the functional effector molecule into cell cytoplasm.

## 2. Results

### 2.1. Preparation of the AuNP Conjugates and Basic Characterization

#### 2.1.1. Detection of Maximum Absorbance Peak

The successful formation of the AuNPs was observed as a color change from yellow (cell-free conditioned media) to crimson (conditioned media challenged with gold salt) (Figure 1A). There was also no color difference among the AuNPs and AuNPs-antimiR 135b and AuNPs-antimiR 135b-Tf conjugates (Figure 1A), indicating the colloid stability of the tested samples. After conjugating AuNPs to different molecules, we observed changes in their maximum absorbance peak (MAP). The spectra pattern regarding peak width or shape was similar in all tested samples (Figure 1B).

As shown in Figure 1B, the MAP of the samples has shifted slightly. The AuNPs had a maximum absorbance peak at 528 nm, while the maximum absorbance peaks for AuNPs-antimiR 135b and AuNPs-antimiR 135b-Tf were determined to be 526 nm and 532 nm, respectively. These shifts point to the presence of antimiR 135b and Tf on the surface of the AuNPs. Produced AuNPs and AuNPs-antimiR 135b-Tf remained stable during the whole duration of the study, and the absorbance spectra of long-term (over 12 months) stored samples can be seen in Appendix A.

#### 2.1.2. Electrophoretic Delay

The detection of RNA migration delay in the electrophoretic field was used to confirm the successful conjugation of antimiR 135b to AuNPs. Here, the AuNPs-antimiR 135b migrated in an electric field of 2% agarose gel. In parallel, a free antimiR 135b served as the migration control.

A delay in the electrophoretic migration of the AuNPs-antimiR 135b conjugate compared to the control antimiR 135b was observed (Figure 2). The migration delay of the heavier AuNPs-antimiR 135b confirmed the successful conjugation of AuNPs to antimiR 135b. The electrophoresis was performed in independent triplicates.

#### 2.1.3. Transferrin Detection

The presence of Tf on the surface of the AuNPs in AuNPs-antimiR 135b-Tf was analyzed using LC–MS, with free Tf as a control. Table 1 shows the LC–MS results. Since the peptides have been measured, the extracted ion chromatogram of one selected transferrin peptide, DC(+57.02)HLAQVPSHTVVAR, which was observed in all the triplicates, is shown in Figure 3.

#### 2.1.4. Size, Charge, and Load of the Conjugates

The size of the AuNPs measured with DLS/Zetasizer was 13.04 nm ± 1.33, and the zeta potential was −35.8 mV ± 1.17. The size measured with DLS is referred to as hydrodynamic diameter.

Using standard curves, we estimated that, on average, AuNP conjugates contained 0.91 ± 0.02 μmol of antimir 135b and 0.008 ± 0.000 mg/mL of Tf per 1 mg of AuNPs.

As expected, the AuNPs exhibited larger sizes after conjugation to Tf and antimiR 135b, and the most negative zeta potential was observed in the AuNPs-antimiR 135b conjugate. This is likely due to the negative charge of the RNA, which increased the overall negative charge of the AuNPs after the conjugation process. The size and zeta potential of the conjugates and the source AuNPs are shown in Table 2.

### 2.2. Evaluation of the AuNPs Uptake

#### 2.2.1. Visualization of the AuNPs Intracellular Location

TEM results confirmed the size of the AuNPs prior to cell culture administration. Most of the AuNPs exhibited sizes between 10 and 16 nm, which agrees with the mean hydrodynamic diameter of the unconjugated AuNPs determined with the Zetasizer. The TEM visualization of the AuNPs and distribution of their sizes are shown in Figure 4.

Subsequently, 4T1 cells without any AuNPs or inhibitors were used as controls, and the representative visualization with basic description of nuclei, mitochondrion, and autophagosomes is shown in Figure 5.

The TEM imaging of the 4T1 cells incubated with the AuNPs conjugates showed different stages and means of the internalization of the AuNPs-antimiR 135b and the AuNPs-antimiR 135b-Tf (Figure 6). We could distinguish free AuNPs-antimiR 135b and identify Intraluminal vesicles (ILVs) in the early endosome and the presence of AuNPs-antimiR 135b inside the early and late endosomes. Some of the AuNPs seemed to be attached to the cell surface, likely preceding internalization. Microvilli and the engulfment of AuNPs-antimiR 135b via macropinocytosis was also detected, as well as the formation of endocytosis vacuole. Representative images proving the above-described observations are shown in Figure 6.

The internalization of the AuNPs-antimiR 135b was then observed in 4T1 cells pretreated with nystatin. Here, we observed the internalization of the AuNPs into early and late endosomes, as well as present microvilli pointing out the engulfment of the conjugate via macropinocytosis. These observations are presented in representative cells in Figure 7.

The 4T1 cells pretreated with chlorpromazine were incubated with both conjugates (i.e., AuNPs-antimiR 135b or AuNPs-antimiR 135b-Tf) because chlorpromazine blocks Tf-mediated endocytosis. The pattern was similar in both types of samples: we identified free conjugates as well as conjugates in early and late endosomes. We also observed the formation of microvilli and the engulfment of conjugates via macropinocytosis (Figure 8).

In the 4T1 cells pretreated with amiloride hydrochloride, the AuNPs-antimiR 135b was detected only free, attached to the cell surface, or in early endosomes (Figure 9). In Figure 9A, we suggested possible internalization of the AuNPs-antimiR 135b by endocytosis that likely follows the attachment of the conjugates to the cell surface (pink/black arrows).

To confirm the presence of the AuNPs-antimiR 135b inside the cells observed by TEM, we employed an EDS spectrum analysis of two different locations (with and without observed AuNPs-antimiR 135b signals). The sample used here was represented by the 4T1 cells incubated with the AuNPs-antimiR 135b without any chemical pretreatments. The EDS analysis confirmed the presence of elemental Au in the location with the AuNPs-antimiR 135b signal, while no Au was detected in the background (Figure 10).

The uptake analysis presented in Figure 5, Figure 6, Figure 7, Figure 8 and Figure 9 shows that the AuNPs-antimiR 135b and/or AuNPs-antimiR 135b-Tf can enter the cells through different combinations of mechanisms.

Interestingly, the AuNPs-antimiR 135b were not observed inside the nuclei, which may be attributed to the short incubation time (2 h) and the immediate fixation of the cells. Alternatively, it may be as in our previous study, where AuNPs were not found inside over 2000 nuclei of breast tumor tissue in mice using TEM [3]. Most nanoparticles were found in early endosomes containing intraluminal vesicles (ILVs). However, ILVs were not observed in the early endosomes of cells treated with amiloride hydrochloride (Figure 9A,B). The AuNPs-antimiR 135b were not agglomerated and were also observed in late endosomes, as well as free particles in the cytoplasm of the cells. The presence of microvilli and the attachment of AuNPs-antimiR 135b or AuNPs-antimiR 135b-Tf to the microvilli were evident in all the images, except of cells treated with amiloride hydrochloride. For instance, in Figure 6C and Figure 7A, the attachment of AuNPs-antimiR 135b to the cell surface structure is visible alongside microvilli, which likely aids in the internalization of the AuNPs-antimiR 135b. The presence of such structures is not evident in the corresponding image in Figure 6D, indicating that different internalization methods are simultaneously at play. An example of internalization is shown clearly in Figure 6C (pink arrow), where it appears that the next step after attachment to the specific structure of the cell membrane is the engulfment of the AuNPs-antimiR 135b, leading to the formation of endocytosis vacuole (purple arrow). Endocytosis without the presence of microvilli for the engulfment of the AuNPs-antimiR 135b can be seen in Figure 9A, where the AuNPs-antimiR 135b are directly located inside the early endosomes (black arrows).

#### 2.2.2. Quantitative Analysis of Internalized AuNPs

To compare the uptake efficacy of different AuNP conjugates, we first counted the number of AuNPs per cell using the TEM at a magnification of 500 nm at several different locations within the randomly acquired cells. The total number of AuNPs in three representative cells is shown in Table 3.

The TEM-based analysis of internalized AuNP conjugates showed no significant differences in total amounts of AuNPs (*p*-value > 0.05) between the two control groups (i.e., AuNPs-antimiR 135b and AuNPs-antimiR 135b-Tf), which indicates that clathrin-mediated endocytosis, caveolin-mediated, and macropinocytosis pathways had similar effects on AuNPs’ cell uptake (Table 3). After treating the cells with different chemicals, it is evident that, when the clathrin-mediated endocytosis pathway is blocked (using chlorpromazine), the highest amount of the AuNPs-antimiR 135b is observed inside the cells, confirming that clathrin-mediated endocytosis is not the main mechanism of the AuNPs’ cell uptake (Table 3). Blocking either the caveolin-mediated or macropinocytosis pathways, both led to a lower amount of the internalized AuNPs (Table 3). Moreover, AuNPs were predominantly observed in vesicular structures such as early endosomes and late endosomes/lysosomes across all tested groups (Table 3). Although TEM shows the presence of the AuNPs within vesicle structures, some free AuNPs were also observed inside the cell cytoplasm. Thus, the short incubation time (2 h) used in this uptake study is sufficient for the AuNPs’ internalization and commencing the release of the AuNPs into cytoplasm. Prolonged incubation time will likely enhance the number of AuNPs in cytoplasm.

### 2.3. Efficacy of the Cargo Delivery by the AuNPs

#### 2.3.1. Detection of the miR 135b Knock-Down

Since we detected effective internalization of the AuNPs by 4T1 cells by all the tested pathways, we wondered if there is any difference in the cargo functionality. Our cargo antimir 135b is an inhibitory short RNA targeting miR 135b overexpressed in the 4T1 cells. Once a functional antimir 135b is released into cell cytoplasm and pairs with target miR 135b, it triggers its degradation. Thus, the efficacy of the AuNPs’ internalization and functionality of the cargo can both be detected by the qPCR of the intracellular miR 135b levels (Figure 11). The level of miR 135b (arbitrary units) shown in Figure 11 represents the fold of change in the miR 135 b level with respect to a reference value detected in non-treated cells.

The miR 135b level in the cells incubated with HP showed borderline downregulation based on the 0.5 cut-off (normalized level = 0.502 ± 0.13), which is probably due to the low incubation time of 2 h. On the other hand, conjugates AuNPs-antimiR 135b (normalized level = 0.17 ± 0.03) and AuNPs-antimiR 135b-Tf (normalized level = 0.44 ± 0.03) showed a remarkable inhibition of miR 135b despite the short incubation time. This shows promising fast uptake and high efficacy of cargo delivery by the AuNPs.

The caveolin-mediated endocytosis and macropinocytosis seem likely to be the main mechanisms for the AuNP-antimiR 135b internalization since their blockage by nystatin and amiloride, respectively, did not result in remarkable miR 135b decrease. The chemicals alone did not affect the level of miR 135b either.

The 4T1 cells pretreated with chlorpromazine exhibited a significant difference in the level of miR 135b compared to the negative control (*p* < 0.01). This indicates that blocking the clathrin-mediated endocytosis pathway does not fully prevent the internalization of the AuNPs. Blocking clathrin-mediated endocytosis allows AuNPs to be internalized through other methods. Also, the Tf-driven enhancement of the AuNPs-antimiR 135b-Tf conjugate uptake did not seem to be remarkably affected by the chlorpromazine treatment. And, in both cases (AuNPs-antimiR 135b and AuNPs-antimiR 135b-Tf), the antimiR 135b cargo remained active.

#### 2.3.2. Quantitation of the AuNPs Uptake with Respect to the Cargo Efficacy

If we consider that each AuNP can carry a similar amount of the antimiR, then the decreased level of target miR 135b roughly reflects the number of conjugates in cytoplasm. The efficacy of the miR 135b knock-down was calculated as ratios of normalized miR 135b levels in samples and non-treated cells (Table 4). The best efficacy in miR 135b knock-down was achieved using the AuNPs-antimiR 135b conjugate, followed by the AuNPs-antimiR 135b-Tf and X-tremeGENE™ HP DNA Transfection Reagent. Interestingly, the chlorpromazine treatment resulted in a lower efficacy of the AuNPs-antimiR 135b conjugate (83% versus 49%) but its effect on the AuNPs-antimiR 135b-Tf had a slightly adverse trend—54% versus 64%. Increased efficacy in the AuNPs-antimiR 135b was also detected after amiloride treatment (18%). However, the effects of chlorpromazine and amiloride on the AuNPs-antimiR 135b are borderline due to higher SDs.

A comparison of the miR 135b knock-down among non-treated cells and those differently inhibited internalization pathways showed that cells pretreated with amiloride hydrochloride or nystatin exhibited a similar miR 135b level as control non-treated cells. Opposingly, the cells pretreated with chlorpromazine showed effective miR 135b inhibition, which means that the AuNPs-antimiR 135b effectively entered the cells via alternative, non-blocked pathways. Thus, clathrin-independent pathways such as caveolin-mediated endocytosis and macropinocytosis are the primary methods of the AuNPs’ internalization, delivering functional cargo into cell cytoplasm.

## 3. Discussion

After attachment to different molecules, AuNPs can exhibit differences in patterns of their spectra and shifts in maximum absorption peaks (MAPs). The surface plasmon resonance (SPR) shift of the AuNPs after attachment to different cargoes is known to occur due to changes in their size, charge, or polydispersity [14]. It has been reported that an increase in the size of nanoparticles corresponds to a shift towards higher wavelength. Agglomeration, polydispersity, and the production of larger shapes result in a significant shift, exceeding 550 nm [6].

The conjugation of the antimir 135b and/or Tf increased hydrodynamic diameter in comparison to source AuNPs, which was expected. Importantly, the zeta potential of the conjugates remained under −30 mV, which is important regarding conjugate stability.

Regarding the NB-AuNPs, there are studies demonstrating that different chemical and physical properties of the nanoparticles such as size, charge, shape, and coatings are responsible for their effect on cell internalization. On the other hand, there are no such data available on the AuNPs with potential use for gene therapy or drug delivery. The activity and the aforementioned properties of the AuNPs are altered by physiological conditions and the formation of a protein corona on the surface of the nanoparticles. In the current study, we used 4T1 cells as a model, since these cells express high amounts of miR 135b, and previous extensive studies on this model allow us a more complex overview [3,11,12]. We are aware that, in different cell lines, the uptake of the AuNPs may occur via different internalization pathways, and a general statement cannot be made without more extensive studies. The current study is the first report on the uptake of the AuNPs and the determination of their internalization pathway in 4T1 cells. The murine 4T1 cells reflect well the behavior of Stage IV, Triple-Negative human breast cancer, including target metastatic sites [15]. Given the similarities with human pathology, our findings might be later applicable in drug development for human use.

As demonstrated previously, different cell types vary in their uptake of the AuNPs [11], and varying concentrations of plasma will result in different protein corona compositions on the surface of the AuNPs [4]. Therefore, in order to mimic the same in vivo conditions, we utilized FBS in the cell culture to simulate the effects of the protein corona on the internalization of the AuNPs.

Regarding active cargo delivery, it was reported that the NB-AuNPs internalized through the clathrin-dependent pathway were destined for lysosomal degradation, while those internalized through other pathways (i.e., clathrin-independent pathways) were packaged in endosomes or caveosomes and sorted to a non-degradative pathway [16,17,18,19]. Using bio-produced AuNPs, we observed some internalized AuNPs in 4T1 cells pretreated with all three endocytosis inhibitors, although the quantity of internalized NPs varied. Important information was obtained studying the cargo functionality, which showed if the particular internalization pathway resulted in successful cargo delivery into cytoplasm. When the clathrin-dependent pathway was blocked, the AuNPs-antimiR 135b could enter via other simultaneous pathways. If the clathrin-dependent pathway is not blocked and the other pathways were blocked, the AuNPs-antimiR 135b or AuNPs-antimiR 135b-Tf were internalized through the clathrin-dependent pathway, and the antimiRs effects were likely diminished due to lysosomal digestion, as suggested in above-mentioned reports on NB-AuNPs. While the studies on NB-AuNPs reported that clathrin-mediated endocytosis is the primary method of NB-AuNPs’ internalization, our findings indicated that bio-produced AuNPs utilized both clathrin-dependent and -independent endocytosis pathways. However, a greater load of active cargo was delivered to the cells via clathrin-independent endocytosis. This difference may be due to the source of the coating on the surface of the AuNPs, which influences their cellular uptake and fate [20].

Receptor-mediated endocytosis—specifically, clathrin- and caveolae-coated vesicles—is known to be the most prevalent method of NB-AuNP internalization [21,22]. Different in vitro studies have confirmed that the shape, size, and other physical properties such as the surface coating of NB-AuNPs determine the type of endocytosis. For example, clathrin-mediated endocytosis has been reported for Tf-coated NB-AuNPs of varying sizes, while 16 nm polyethylene glycol (PEG)-coated NB-AuNPs showed both clathrin- and caveolae-mediated endocytosis [23]. In the case of 4.5 nm PEG-coated NB-AuNPs, only caveolae-mediated endocytosis was observed [24], and, for 20 nm Fetal Bovine Serum (FBS)-coated NB-AuNPs, clathrin-mediated endocytosis was reported [20]. The capping agent (coating) of the bio-produced AuNPs, which is secreted by the fungal cells, is an important factor for their internalization method, as well as for releasing the active cargo into the cell cytoplasm. The capping agent was shown to involve peptides and/or amino acids on the surface of the AuNPs, as reported in our previous study [4]. There is a possibility of endosomal destabilization by a particular capping agent, promoting the cytoplasmic release of the cargo. The capping agents from various producers can also differently mask physical–chemical properties of the AuNPs core. There is a further need to study AuNPs with different physical properties (such as size, shape, zeta potential, etc.) to demonstrate and confirm their similarities in internalization with the NB-AuNPs. Even though the short antimiR sequences are generally more stable than longer RNA oligos, the AuNPs likely function as a further protection of ssRNAs [12,25] and hold promise as a safe and effective carrier for gene therapy. The AuNPs-antimiR 135b were not observed inside the nuclei, which may be attributed to the short incubation time (2 h) and the immediate fixation of the cells, or it is indeed the fate of the AuNP carriers to stay in cytoplasmic pools. This observation agrees with our previous study, where AuNPs were not found inside over 2000 nuclei of breast tumor tissue in mice using TEM [3]. The long-term persistence of the AuNPs within cytoplasm and its subsequent effect on the particular cell still needs to be clarified.

Since the uptake studies are usually performed with the aim to prove that the carriers can deliver a particular cargo (i.e., drug) into cells, it is important to focus not only on the mean of uptake but also on the functionality of the carried molecule by a different approach. In the case of carriers for RNA-based therapy, used in this study, a parallel examination with microscopy (TEM) and functional study (qPCR) gave us valuable information.

The methodology for NPs’ uptake, even in the case of more studied NB-AuNPs, is not unified, and every type of experimental focus or need for particular outputs has its own approach. For example, Dosumu et al. applied a coating of a red-luminescent ruthenium transition metal complex on the surface of NB-AuNPs and tracked the luminescence signal of the NB-AuNPs inside untreated A549 cells using confocal microscopy. They used a combination of Inductively Coupled Plasma Mass Spectrometry (ICP–MS) and TEM to complete their investigations [26]. In another study, Yilmaz et al. utilized various endocytosis inhibitors against three cell lines (Beas-2b, A549, and PNT1A) and compared the internalization pathways of NB-AuNPs using two label-free methods: flow cytometry (due to the side scatter shift of the cells after AuNP internalization) and surface-enhanced Raman spectroscopy (SERS, by comparing different spectra) in the presence of controls [16]. It was shown that the clathrin-mediated endocytosis of NB-AuNPs occurred in the presence of endocytosis blockers in two different cell lines (MRC5 lung fibroblasts and Chang liver cells), which was confirmed by TEM and Scanning Electron Microscopy (SEM) techniques, and quantified by ICP–MS and verified by confocal microscopy [20]. We omitted the microscopy techniques such as fluorescent labeling for tracking the AuNPs due to issues with photobleaching, as well as the possibility that their attachment to the AuNPs may alter the surface charge, coating, size, and other properties of the AuNPs, which could affect their true route of internalization and cargo release. Also, the AuNPs possess some autofluorescence that may distort the actual results. Other available techniques, such as ICP–MS or GF-AAS, were not used because their detection limit for AuNPs is high. Instead, we employed TEM and qPCR as two different approaches that were the most suitable for our research focus (internalization and functionality of delivered cargo).

Using chemical inhibitors to study different internalization pathways may trigger off-target effects and potential cytotoxicity of the chemical inhibitors. In the current study, we employed three different chemical treatments: nystatin at a final concentration of 50 µM, amiloride hydrochloride hydrate at 0.5 mM, and chlorpromazine hydrochloride at 0.03 µM. The concentrations used for each chemical were below their respective cytotoxic levels [17,18]. Two of the chemicals mentioned above (nystatin and amiloride hydrochloride) exhibit known off-target activities. For instance, nystatin is primarily used as an antifungal agent, but it also exhibits off-target effects such as binding to cholesterol, disrupting cholesterol-enriched membrane microdomains (lipid rafts) and altering internalization pathways [27]. We have leveraged this off-target activity of nystatin for our current research. Other off-target effects of nystatin, including nephrotoxicity, pro-inflammatory responses, and impacts on axon growth and regeneration, are observed mainly in vivo and are not relevant to our in vitro study. Amiloride hydrochloride hydrate is a pyridine compound used to treat hypertension and congestive heart failure. It functions as a potassium-sparing diuretic by blocking epithelial sodium channels (ENaC) in the distal nephron of the kidneys. Additionally, amiloride’s inhibition of Na^+^/H^+^ exchange may indirectly influence macropinocytosis by altering sub-membranous pH, which is essential for the activity of GTPases involved in actin remodeling—a critical step in macropinosome formation [28]. While its primary mechanism of action as a drug is the modulation of renal ion channels, amiloride can also inhibit macropinocytosis. This effect is likely due to changes in sub-membranous pH resulting from its inhibition of Na+/H+ exchange, representing an off-target action that we leveraged in our study.

Another chemical we used in our study was chlorpromazine hydrochloride, an antipsychotic medication primarily used to treat psychotic disorders such as schizophrenia. As a cationic amphiphilic drug, chlorpromazine exerts its effects by binding to clathrin and AP2, thereby inhibiting their proper assembly into clathrin-coated vesicles at the plasma membrane. Additionally, it may impede dynamin, a GTPase essential for clathrin-mediated endocytosis. Chlorpromazine has also been shown to disrupt basolateral actin cytoskeleton dynamics, which are important for AQP2 endocytosis in kidney epithelial cells. However, detailed studies on the off-target effects of this drug are limited [29].

## 4. Materials and Methods

### 4.1. AuNPs’ Source and Preparation

The AuNPs were produced by *F. oxysporum*, according to our established protocol [3]. Briefly, *F. oxysporum* was cultured in Sabouraud Dextrose Broth (SDB, Sigma-Aldrich, Prague, Czech Republic) at 30 °C for one week, and the cell culture supernatant was used for AuNP production in the presence of a final concentration of 1 mmol HAuCl_4_·3H_2_O (Sigma-Aldrich, Prague, Czech Republic). After heating the supernatant at 80 °C for 5 min, in the presence of a control sample (i.e., SDB containing a final concentration of 1 mmol HAuCl_4_·3H_2_O), AuNPs were washed three times with ddH_2_O and collected by centrifugation at 15,000× *g* for 30 min.

To characterize the size and charge of the AuNPs, we employed dynamic light scattering (DLS) through a non-invasive backscattering method and electrophoretic light scattering (ELS), respectively, with a Zetasizer Ultra instrument (Malvern Panalytical, Malvern, UK). The measurements employed a refractive index of 0.18 and an absorbance of 3.43, and utilized double-distilled water (ddH_2_O) as the dispersant. For size distribution assessment, an ultra-low-volume quartz cuvette (ZEN2112, Malvern Panalytical) was used, with data collected in backscatter mode. Zeta potential measurements were performed using a DTS1070 zeta cell with folded capillaries, all at a controlled temperature of 25 °C [30]. Samples were measured in triplicates.

### 4.2. Preparation of AuNPs Conjugates

#### 4.2.1. AuNP-AntimiR 135b and AuNP-AntimiR 135b-Tf Preparation

To prepare AuNP-antimiR 135b, the used sequence was antisense microRNA 135b with phosphorothioate bonds (PS) (5′ UpsCpsAps CAU AGG AAU GAA AAG CCpsAps UpsA 3′, Sigma Aldrich, Prague, Czech Republic) and followed optimized protocol reported earlier [3]. Briefly, 200 µL of the washed and sterilized AuNPs (1.89 ± 0.08 mg/mL, according to our previous graphite furnace atomic absorption spectrometry (GF-AAS) data [3]) was added to 100 µL of antimiR 135b (100 µmol), and the conjugate was incubated in a thermomixer at 1200 rpm at 4 °C for 24 h. AuNP-antimiR 135b-Tf was prepared in the same manner with the following amounts: 200 µL of AuNPs with 100 µL of antimiR 135b (100 µmol) and 5.4 µL of Tf (5 mg/mL, Life Technologies, Prague, Czech Republic). The samples were washed three times using RNase-free ddH_2_O and centrifugation at 15,000 rcf for 30 min and resuspended in 200 µL of RNase-free ddH_2_O. In order to ensure that the unbound antimiR 135b and Tf were washed from the sample, a Tecan spectrophotometer (Infinite 200 PRO UV-visible spectrophotometer, Tecan, Männedorf, Switzerland) was used and the fluorescence intensity of the supernatant from the final washing step was determined at excitation/emission wavelengths of 488/520 nm and 580/609 nm for determining the amounts of unbound antimir 135b and Tf, respectively. The free and diluted amounts of antimir 135b (1 µmol) and Tf (0.5 mg/mL) were used as positive controls [3], and each measurement was conducted three times.

Determining the presence of antimiR 135b and Tf on the surface of the AuNPs, their amounts and conjugate characterizations involved using various methods, which are discussed below.

#### 4.2.2. Absorption Spectra Measurement

After loading different molecules onto the AuNPs, we tested the spectra pattern and shift of maximum absorption peaks (MAPs) in comparison to the original AuNPs. An increase in AuNPs’ size, pointing to, for example, agglomeration, results in a significant spectrum shift towards higher wavelength. To see spectra patterns and their shifts after the formation of conjugates, we used the Tecan spectrophotometer with 1 nm step resolution and measured wavelength from 400 to 650 nm. The blank consisted of ddH_2_O used for the conjugate preparation and the measurements were conducted in triplicate.

#### 4.2.3. Size and Surface Charge Characterization

To characterize the size and charge of the AuNP conjugates, triplicates of test samples (AuNPs-antimiR 135b and AuNPs-antimiR 135b-Tf), along with control triplicates (AuNPs), were assessed for their size and zeta potential using DLS and ELS with the Zetasizer, as detailed in Section 4.1.

#### 4.2.4. Tf and AntimiR Amount Calculations

First, standard curves of the known amounts of each molecule (Tf and antimiR 135b) were prepared based on the detection of specific excitation/emission wavelengths using a Tecan spectrophotometer, as described previously [3]. The supernatant of the conjugate was analyzed prior to the addition of water and washing of the pellet, and the number of unbound molecules was determined using the standard curves. Finally, the number of bound molecules was calculated using the equations from each curve and an online tool available at www.wolframalpha.com (accessed on 7 September 2025) [3]. Tests were conducted in triplicate.

#### 4.2.5. Proof of AntimiR Conjugation by Electrophoretic Mobility

The conjugation of AuNPs to antimiR 135b changes the electrophoretic mobility of the antimiR 135b due to the heavier nature of the conjugate [3,12]. The 2% *w*/*v* agarose gel (Sigma Aldrich, Prague, Czech Republic) plus GelRed™ nucleic acid stain (Biotium, CA, USA) was prepared in Tris–acetate–EDTA buffer (TAE) and the samples (i.e., antimiR 135b and AuNPs-antimiR 135b) were mixed with MassRuler DNA loading dye (Thermo Fisher Scientific, Waltham, MA, USA) and loaded into the wells. One well was filled with MassRuler DNA ladder (Thermo Fisher Scientific, Prague, Czech Republic). The voltage used was 110 V and it ran for 45 min in TAE buffer [12]. The gel and RNA bands were observed using a gel documentation system, and electrophoresis was repeated twice to confirm the results.

#### 4.2.6. Proof of Tf Conjugation by a Liquid Chromatography–Mass Spectrometry (LC–MS)

Three independently prepared samples of AuNPs-antimiR 135b-Tf were collected and analyzed by LC–MS. Samples (AuNPs-antimiR 135b-Tf as test and Tf as control) were combined with 30 µL of 50 mM ammonium bicarbonate along with dithiothreitol (DTT) at a final concentration of 10 mM. This mixture was then incubated at 60 °C for 40 min. Once allowed to cool to room temperature, iodoacetamide was introduced to reach a final concentration of 30 mM, and the samples were incubated in darkness for 30 min. To halt the alkylation process, additional DTT was added to achieve a final concentration of 50 mM. Subsequently, trypsin was incorporated to a concentration of 0.1 µg/mL, and the mixture was allowed to incubate overnight at 37 °C. The resulting samples were subjected to analysis using a liquid chromatography system (Agilent 1200 series, Agilent Technologies, Santa Clara, CA, USA) linked to the timsToF Pro PASEF mass spectrometer, which featured a Captive spray (Bruker Daltonics, Fremont, CA, USA) operating in positive data-dependent mode. An autosampler injected 5 µL of the peptide mixture into a C18 trap column (UHPLC Fully Porous Polar C18 2.1 mm ID, Phenomenex, c/o Beckman Coulter, Prague, Czechia). After 5 min of trapping at a flow rate of 20 µL/min, the peptides were separated on a C18 column (Luna Omega 3 μm Polar C18 100 Å, 150 × 0.3 mm, Phenomenex) using a linear water–acetonitrile gradient over 35 min, transitioning from 5% (*v*/*v*) to 35% (*v*/*v*) acetonitrile at a flow rate of 4 µL/min. Both the trap and analytical columns were maintained at 50 °C.

For the timsTOF Pro settings, standard proteomics parameters from the PASEF method were applied. Specifically, the target intensity for each individual PASEF precursor was set at 6000, with an intensity threshold of 1500. The scan range was configured between 0.6 and 1.6 V s/cm^2^, with a ramp time of 100 ms. A total of 10 PASEF MS/MS scans were conducted. Precursor ions within an *m*/*z* range of 100 to 1700 with charge states ranging from ≥2+ to ≤6+ were selected for fragmentation, with active exclusion implemented for 0.4 min.

Raw data processing was carried out using PeaksStudio 10.0 software (Bioinformatics Solutions, Waterloo, ON, Canada). The search parameters included trypsin as the enzyme (specific), carbamidomethylation as a fixed modification, and the oxidation of methionine along with the acetylation of the protein N-terminus as variable modifications. The database utilized for searches was UniProt (all taxa, November 2021).

### 4.3. Cell Culture and Blocker Dosages

The 4T1 mouse mammary adenocarcinoma (ATCC CRL-2539) was used in this study [11], and a working medium, consisting of Roswell Park Memorial Institute Medium 1640 (RPMI-1640, Sigma Aldrich, Prague, Czech Republic), 10% fetal bovine serum (FBS, Gibco, MA, USA), 4.5 g/L glucose (Sigma Aldrich, Prague, Czech Republic), and 44 µg/mL gentamicin (Sandoz, Novartis Company, Prague, Czech Republic), was used for culturing the cells. The MTT assay for the toxicity assessment of the AuNPs was not conducted according to the IC_50_ determined from the previous study, which showed that AuNPs with an initial dose of 61.1 µg/100 µL did not induce toxicity higher than the IC_50_ in the cell culture [3]. Therefore, the same amounts were used in this experiment.

In order to block the caveolin-mediated pathway, we used nystatin (0.5 mg/mL, Sigma Aldrich, Prague, Czech Republic) at a final concentration of 50 µM [31]. For blocking macropinocytosis, we used amiloride hydrochloride hydrate (10 mM, Sigma Aldrich, Prague, Czech Republic) at a final concertation of 0.5 mM [32]. For blocking the clathrin-mediated pathway, we used chlorpromazine hydrochloride (1 mg/mL, Sigma Aldrich, Prague, Czech Republic) at a final concertation of 0.03 µM [32].

### 4.4. TEM and Energy Dispersive X-Ray Spectroscopy (EDS) Analyses

Additionally, 4T1 cells (2 × 10 ^3^ cells/mL) were cultured in the same working medium as mentioned above, but on round cover slips (12 mm) in a 24-well plate. After 48 h of incubation, the cells reached 60–70% confluence in 500 µL of fresh culture medium. The wells containing cover slips were then divided into 7 groups (each group in triplicate). Two groups were treated with the above-mentioned amounts of nystatin or amiloride hydrochloride, and two groups after treatment with chlorpromazine. After 30 min of incubation, the chemicals were washed twice with PBS, and the wells were refilled with 500 µL of the working medium. In parallel, 3 groups (each in triplicate) remained untreated and served as controls. After pretreatment with chemicals, two groups (nystatin and amiloride hydrochloride) were incubated with 2.5 µL of AuNPs-antimiR 135b. The third and fourth groups (chlorpromazine treatment) were incubated either with AuNP-antimiR 135b or AuNP-antimiR 135b-Tf. The control groups included cells without any treatment, and two groups incubated with 2.5 µL of AuNP-antimiR 135b or AuNP-antimiR 135b-Tf. The incubation time with the AuNPs was 2 h; then, cells were washed with PBS and fixed with 1% glutaraldehyde (GA) containing 2.5% Paraformaldehyde/Sodium cacodylate buffer (PFA/SB buffer). After 1 h of fixation at 4 °C and three washings with SB buffer, 1% OsO_4_ in SB was added, followed by another hour of incubation in the dark at room temperature. The plate was then washed with SB buffer, rinsed with water three times, and incubated with various concentrations of acetone in water (30%, 50%, 70%, 90%, and 95%) before finally using water-free acetone. The samples were embedded and polymerized using Epon-Durcupan. The resin blocks were cut into 85 nm slices using an ultramicrotome (Leica EM UC6, Prague, Czech Republic) and placed on copper grids (square, 200 mesh, Agar Scientific, Essex, UK). Images were acquired using a transmission electron microscope Jeol JEM 1400 Flash (120 kV, JEOL, Ltd., Tokyo, Japan), operated at 80 kV, and equipped with a tungsten cathode and bottom-mounted FLASH 2kx2k CMOS camera (TVIPS GmbH, Gilching, Germany). The same AuNPs used for cell experiments were also analyzed alone using TEM. For this purpose, 2 mL of the sample was placed on a glow-discharge-activated grid (30 s, 1 kV, 10 mA). After air drying, the samples were analyzed by TEM (JEOL JEM-F200), operated at 200 kV and equipped with a Cold FEG and TVIPS XF416 camera (TVIPS GmbH).

To prove the presence of elemental Au in the samples, the control cells incubated with AuNPs were checked using a JED 2300 X-ray spectrometer (JEOL, Ltd.) to obtain EDS spectra from the zones containing AuNPs. The spectra were then analyzed using Jeol AnalysisStation software JEM-F200.

TEM photomicrographs were used to roughly evaluate the amount of internalized AuNPs per cell in each group (i.e., AuNPs-antimiR-135b and AuNPs-antimiR-135b-Tf).

For this purpose, the cells (n = 3/group) were first examined at a magnification of 5 µm, and then AuNPs were counted in at least three random areas of each cell at a magnification of 500 nm. The number of AuNPs was recorded, and the average total number of nanoparticles was calculated for each group. Additionally, colocalization and the quantities of AuNPs within early endosomes and late endosomes/lysosomes were assessed. To compare the results, data were compared using analysis of variance (ANOVA) in ANOVA Calculator—One Way ANOVA and Tukey HSD tests available at https://www.statskingdom.com/180Anova1way.html (accessed on 7 September 2025). *p*-values of ≤0.05 were considered as significant.

### 4.5. Quantitation of miR 135b Knock-Down Using qPCR

Independent duplicates of all the samples for qPCR were prepared. Eight plates were cultured with the same amounts of cells (2 × 10^3^ cells/mL) and, after overnight incubation in the cell culture condition, the plates were incubated with different chemicals; one plate incubated with nystatin, one plate with amiloride hydrochloride, and two plates incubated with chlorpromazine. The concentrations used were based on previous studies involving 4T1 cells [31,32]. The incubation time was 60 min, after which the chemicals were removed and cells were washed with PBS. The plates were then refilled with 1 mL of the working medium. In the first two sets (plates treated with nystatin and amiloride hydrochloride), each plate was incubated with 10 µL of AuNPs-antimiR 135b. The plates in the third set (plates treated with chlorpromazine) were each incubated with 10 µL of either AuNP-antimiR 135b or AuNP-antimiR 135b-Tf. Additionally, seven plates were used as controls: two were filled with 10 µL of either AuNP-antimiR 135b or AuNP-antimiR 135b-Tf, one plate was filled with X-tremeGENE™ HP DNA Transfection Reagent (Sigma Aldrich, Prague, Czech Republic and in this article is named HP)-antimiR 135b serving as a positive transfection control, one untreated plate served as a negative control (cell without any pretreatment and no AuNPs conjugates), and three pretreated plates, each containing the same amounts of chemicals (i.e., nystatin, amiloride hydrochloride, and chlorpromazine), were only refilled with the working medium and used as chemical-treated controls. The plates were then incubated under cell culture conditions, washed with PBS after 2 h, refilled with the working medium, and incubated overnight.

MicroRNA was extracted using the High Pure miRNA Isolation Kit (Roche, Prague, Czech Republic). The High-Capacity cDNA Reverse Transcription Kit (Thermo Fisher Scientific, Prague, Czech Republic) was used to generate cDNA with miR 16 (assay ID 000391) and miR 135b (assay ID 002261) RT primers (Thermo Fisher Scientific, Prague, Czech Republic). The level of miR 135b relative to miR 16 (as an internal control) was determined using real-time polymerase chain reaction (qPCR) and an iQ5 Real-Time PCR Detection System (BioRad, Prague, Czech Republic). The qPCR reaction mixture consisted of TaqMan Universal PCR Master Mix (No AmpErase, Thermo Fisher Scientific, Prague, Czech Republic) with an appropriate mix of primers/probes (miR 135b assay ID 002261 and miR 16 assay ID 000391). Each reaction was run in triplicate. Results were analyzed using the iQ5 Optical System Software 2.1 (BioRad, Prague, Czech Republic), and changes in the level of miR 135b were calculated based on the 2^−ddCt^ method. Remarkable change was considered using cut-off = 0.5. Differences with *p*-value  <  0.01 were considered as significant [3,33].

Since we use an optimized ratio and incubation time for loading antimiR onto AuNPs, we expect a similar load of the antimiR per each AuNP. The successful delivery of the antimiR is demonstrated by the effective but not complete knock-down of the target miR 135b after the cells were treated with the AuNPs-antimiR 135b only. Using the non-treated cells as a reference sample, we can quantify the efficacy of miR 135b knock-down reflecting also the number of internalized AuNPs in cell cytoplasm.

## 5. Conclusions

In the current study, we used biological AuNPs of size about 13 nm produced by conditioned media of fungus *F. oxysporum.* We have been studying biochemical properties of these AuNPs, and they seem to be promising drug carriers. In this study, we focused on their internalization into specialized 4T1 cancer while carrying effector non-coding RNA molecule. The TEM analysis showed that the AuNPs-antimiR 135b conjugates can enter 4T1 cells via all the tested pathways (caveolae-mediated endocytosis, clathrin-independent pathways, and macropinocytosis). Only the caveolae-mediated endocytosis and the macropinocytosis of the AuNPs resulted in the delivery of effector antimiR into cytoplasm and allowed its binding to target miR 135b. Once one of these pathways is blocked and the AuNPs-antimiR 135b primarily enter cells through the clathrin-dependent pathway, the effect of antimiR fails, likely due to lysosomal digestion.

## Figures and Tables

**Figure 1 ncrna-11-00066-f001:**
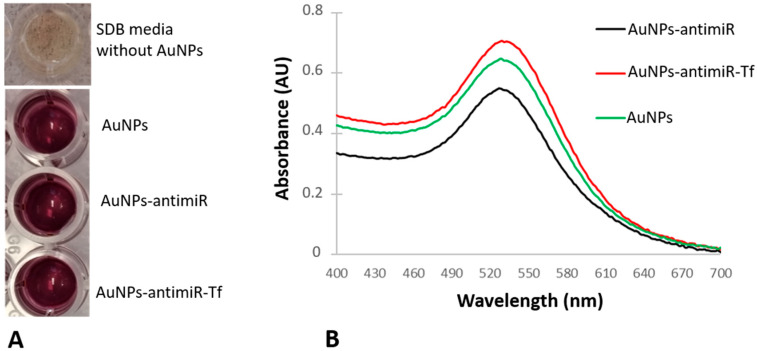
Changes in the color and the maximum absorbance peaks of the conjugates. (**A**) Media color in absence or presence of the AuNPs. (**B**) UV–vis spectra of the original AuNPs and derived conjugates (i.e., AuNPs-antimiR 135b and AuNPs-antimiR 135b-Tf).

**Figure 2 ncrna-11-00066-f002:**
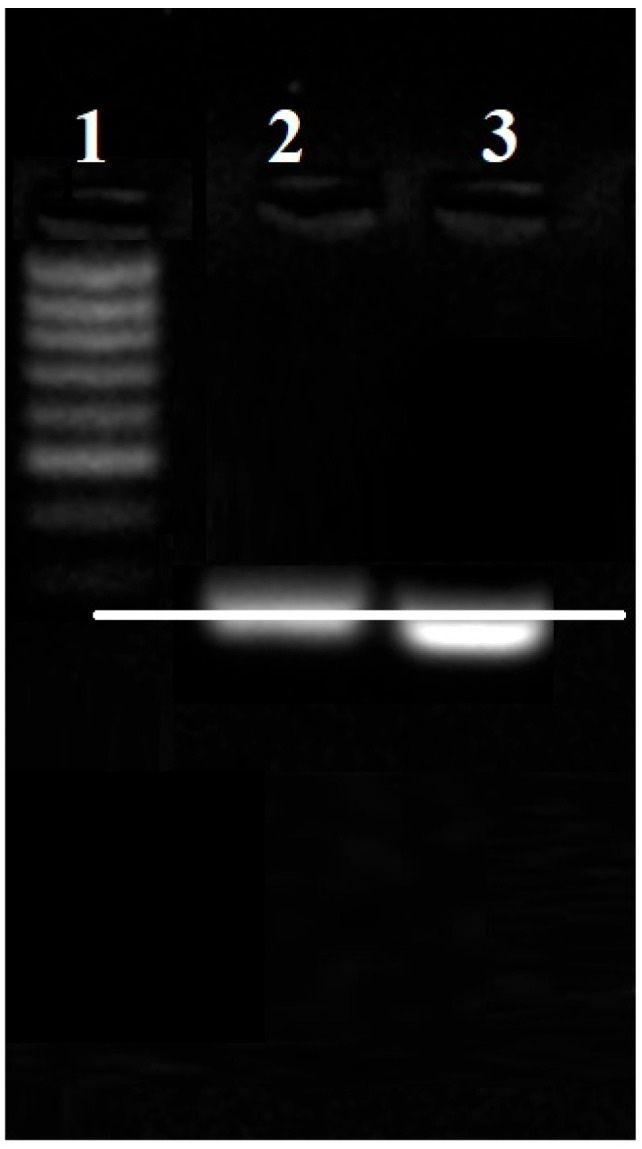
Confirmation of the AuNPs-antimiR 135b conjugation by antimiR migration delay in the electrophoretic field. (1) MassRuler low-range DNA ladder, (2) AuNPs-antimiR 135b, and (3) control-free antimiR 135b. White line points out a lag of migration of AuNPs-antimiR 135b.

**Figure 3 ncrna-11-00066-f003:**
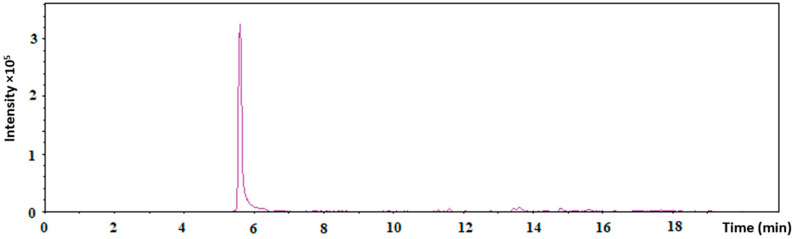
The extracted ion chromatogram of one selected transferrin peptide, DC(+57.02)HLAQVPSHTVVAR, through LC–MS measurements.

**Figure 4 ncrna-11-00066-f004:**
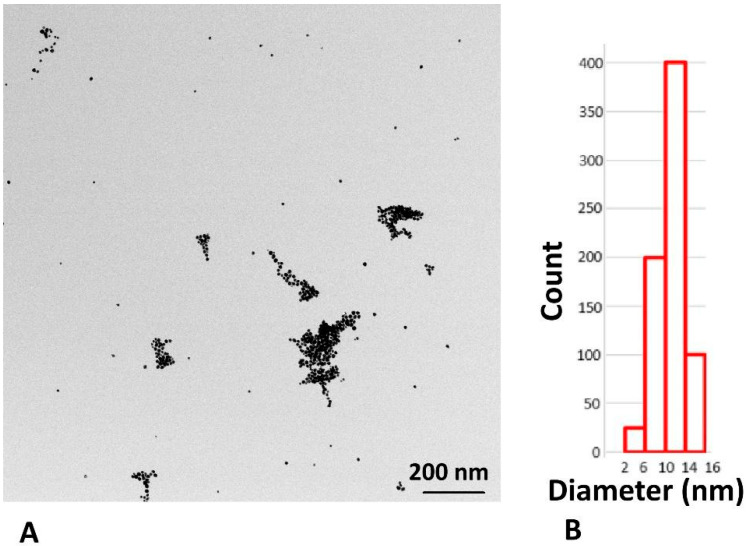
TEM analysis of the washed AuNPs before cell culture application. (**A**) Visualization of the AuNPs by TEM; (**B**) size distribution of the obtained AuNPs.

**Figure 5 ncrna-11-00066-f005:**
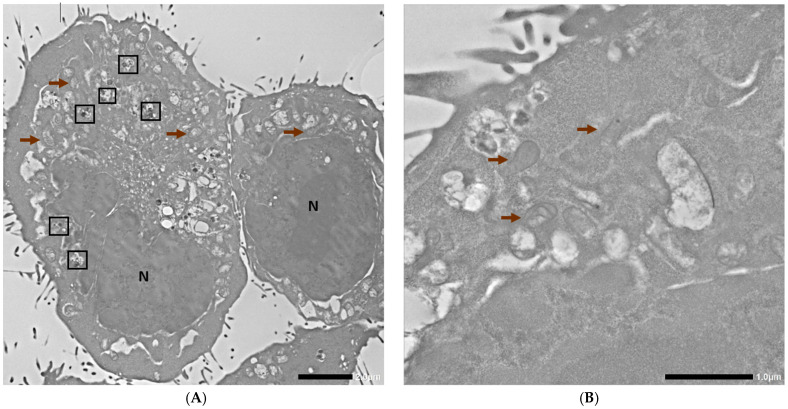
TEM photographs of control 4T1 cells. N corresponds to the nucleus, brown arrows indicate mitochondria, and the black frames indicate autophagosomes. Scale bars are 2 µm and 1 µm for (**A**,**B**), respectively.

**Figure 6 ncrna-11-00066-f006:**
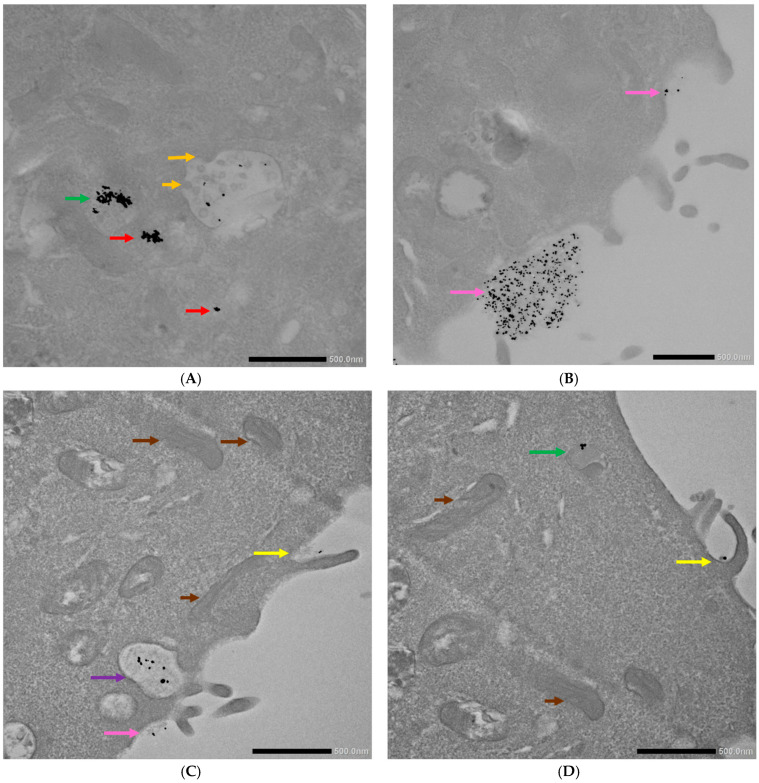
TEM photographs of 4T1 cells incubated with the AuNP conjugates. (**A**,**B**) Display cells after incubation with AuNPs-antimiR 135b and (**C**,**D**) depict cells after incubation with AuNPs-antimiR 135b-Tf. Brown arrows indicate mitochondria; orange arrows indicate ILVs/early endosome; red arrows indicate free conjugates; green arrows indicate late endosomes; pink arrows show conjugates’ attachment to cell surface; yellow arrows indicate microvilli; and purple arrow indicates endocytosis vacuole. Scale bars represent 500 nm.

**Figure 7 ncrna-11-00066-f007:**
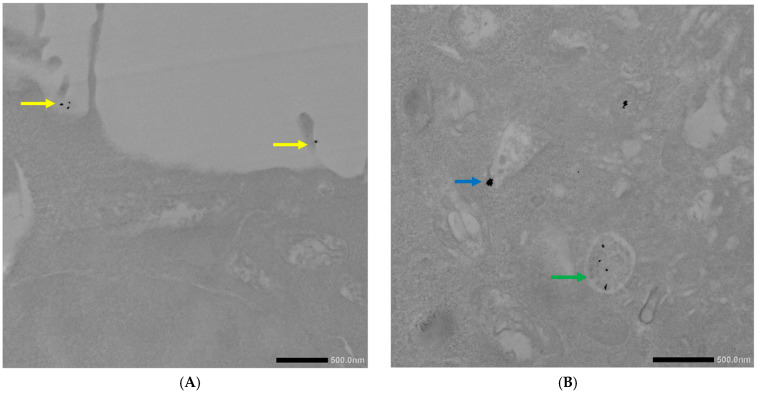
TEM photographs of AuNPs-antimiR 135b uptake by the 4T1 cells pretreated with nystatin. (**A**) Yellow arrows correspond to microvilli and (**B**) blue and green arrows indicate early and late endosomes, respectively. Scale bars represent 500 nm.

**Figure 8 ncrna-11-00066-f008:**
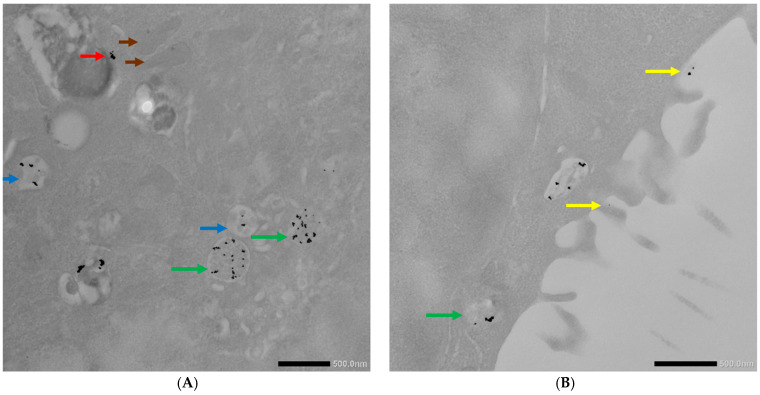
TEM photographs of AuNPs-antimiR 135b or AuNPs-antimiR 135b-Tf uptake by the 4T1 cells pretreated with chlorpromazine. (**A**,**B**) Uptake of the AuNPs-antimiR 135b and (**C**,**D**) uptake of the AuNPs-antimiR 135b-Tf. Brown arrows indicate mitochondria; red arrows indicate free conjugates; blue and green arrows indicate early and late endosomes, respectively; and yellow arrows correspond to microvilli. Scale bars in (**A**–**C**) represent 500 nm and scale bar in (**D**) represents 1 µm.

**Figure 9 ncrna-11-00066-f009:**
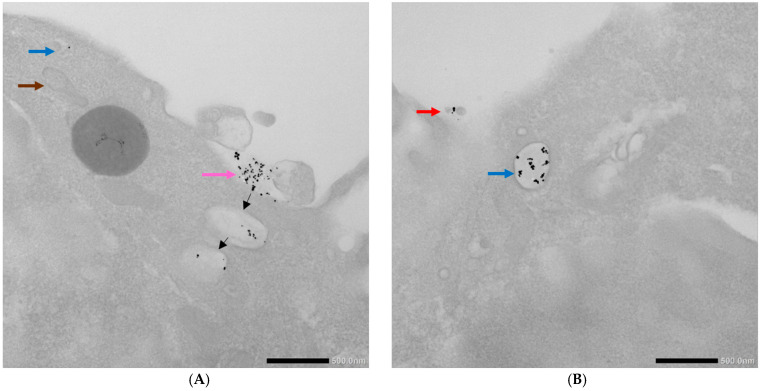
TEM photographs illustrating uptake of AuNPs-antimiR 135b by the 4T1 cells pretreated with amiloride hydrochloride. (**A**,**B**) Brown arrow indicates mitochondrion; red arrow indicates free AuNPs-antimiR 135b; blue arrows indicate early endosomes; and pink arrow shows the surface attachment of the conjugates. Black arrows suggest possible fate of attached conjugates. Scale bars represent 500 nm.

**Figure 10 ncrna-11-00066-f010:**
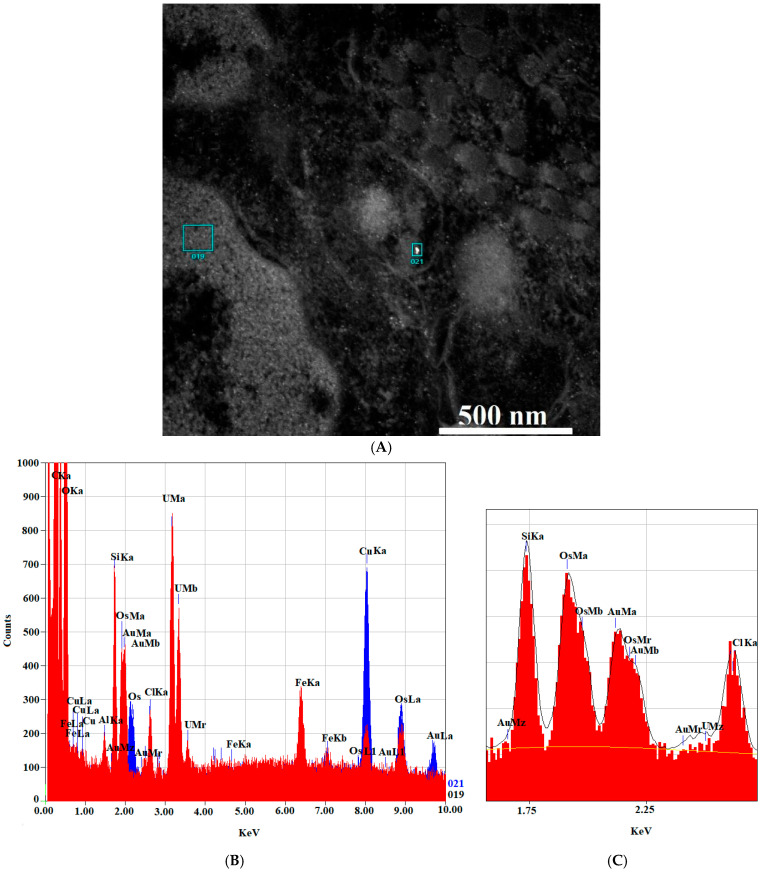
EDS analysis of the AuNPs-antimiR 135b signal. (**A**) The areas chosen for EDS analysis with (area 21) and without (area 19) the AuNPs-antimiR 135b signal. Scale bar is 500 nm. (**B**) EDS spectra showing the AuNPs-antimiR 135b signal (area 21) in red and background (area 19) in blue. (**C**) Shows detailed section of the EDS spectra from 1.5 KeV to 2.5 KeV.

**Figure 11 ncrna-11-00066-f011:**
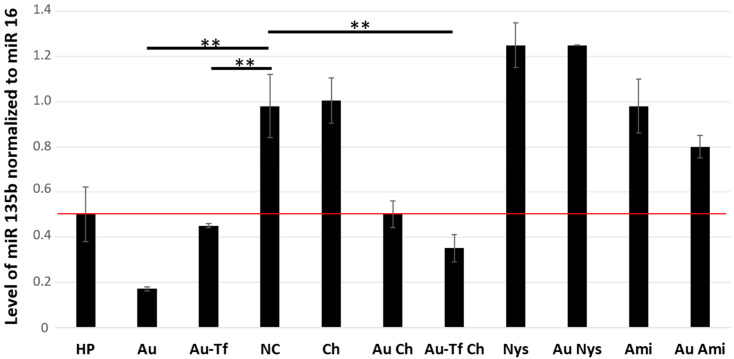
Level of miR 135b in 4T1 cell cytoplasm detected by qPCR. The amounts of miR-135b were normalized to miR 16, used as reference. A fold of change with *p* < 0.01 (**) was considered significant. The cut-off for samples with remarkably decreased miR-135b was fold of change ≤ 0.5, and the red line marks fold of change = 0.5. NC: control, Au: AuNPs-antimiR 135b, HP: X-tremeGENE™ HP DNA Transfection Reagent, Tf: transferrin, Ami: amiloride hydrochloride, Ch: chlorpromazine, and Nys: nystatin.

**Table 1 ncrna-11-00066-t001:** LC–MS results of Tf after conjugation to AuNPs-antimiR 135b.

Protein ID	Accession	Coverage (%)	Avg. Mass	Description
131110	P02787|TRFE_HUMAN	76	77,064	Serotransferrin OS = Homo sapiens OX = 9606 GN = TF PE = 1 SV = 3
131111	A5A6I6|TRFE_PANTR	71	77,064	Serotransferrin OS = Pan troglodytes OX = 9598 GN = TF PE = 2 SV = 1

**Table 2 ncrna-11-00066-t002:** The size (hydrodynamic diameter) and zeta potential of the conjugates and AuNPs measured with Zetasizer (Malvern Panalytical, Malvern, UK). The data were acquired in triplicates.

Tested Material	Size and Zeta Potential	Mean	SD
AuNPs	Size by number (nm)	13.04	±1.33
	Zeta potential (mV)	−35.8	±1.17
AuNPs-antimiR 135b-Tf	Size by number (nm)	67.35	±0.63
	Zeta potential (mV)	−30.48	±0.82
AuNPs-antimiR 135b	Size by number (nm)	32.4	±3.20
	Zeta potential (mV)	−38.84	±1.97

**Table 3 ncrna-11-00066-t003:** Total number of AuNPs in each group compared to the total number located in early endosomes and late endosomes/lysosomes. Nys = nystatin, Ch = Chlorpromazine, Ami = Amiloride hydrochloride.

Total Number of AuNPs in Cells in Each Treatment Group	Cell no 1	Cell no 2	Cell no 3	Mean Number of AuNPs
AuNPs-antimiR 135b	23	140	105	89
AuNPs-antimiR 135b-Tf	25	55	63	48
Nys + AuNPs-antimiR 135b	43	45	32	40
Ch + AuNPs-antimiR 135b	159	56	72	96
Ch + AuNPs-antimiR 135b-Tf	74	19	73	55
Ami + AuNPs-antimiR 135b	48	28	45	40
Total number of AuNPs in early endosomes and late endosomes/lysosomes	Cell no 1	Cell no 2	Cell no 3	Mean number of AuNPs
AuNPs-antimiR 135b	22	105	64	64
AuNPs-antimiR 135b-Tf	25	55	63	48
NY + AuNPs-antimiR 135b	35	21	11	22
CH + AuNPs-antimiR 135b	119	37	62	73
CH + AuNPs-antimiR 135b-Tf	36	16	67	40
AM + AuNPs-antimiR 135b	48	25	41	38

**Table 4 ncrna-11-00066-t004:** Knock-down efficacy of the target miR 135b with respect to non-treated cells. Values are presented as a percentage of target miR 135b level normalized to its reference level in non-treated cells. HP: X-tremeGENE™ HP DNA Transfection Reagent, Ami: amiloride hydrochloride, Ch: chlorpromazine, and Nys: nystatin. Remarkable differences are in bold.

Treatment	Knock-Down Efficacy (%) ± SD
Non-treated cells	0 + 5.0
Positive control (HP)	49 ± 12.2
AuNPs-antimiR 135b	**83 + 1.1**
AuNPs-antimiR 135b-Tf	**54 + 1.0**
Ch	1 + 10.2
Ch + AuNPs-antimiR 135b	49 + 6.0
Ch + AuNPs-antimiR 135b-Tf	**64 + 5.9**
Nys	0 + 10.4
Nys + AuNPs-antimiR 135b	0 + 0.2
Ami	1 + 12.0
Ami + AuNPs-antimiR 135b	18 + 5.1

## Data Availability

All the data and Appendix A associated with this study are presented in this journal paper.

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
