# Peer review of "Cellular Delivery of Functional AntimiR Conjugated to Bio-Produced Gold Nanoparticles"

_ncrna, 2025, doi:10.3390/ncrna11050066_

Round 1
Reviewer 1 Report (Previous Reviewer 1)
Comments and Suggestions for Authors
This is a review of the article " Qualitative cellular tracking of antimiR conjugated bio-produced gold nanoparticles " after minor revision. The authors have incorporated my comments into the submitted manuscript. Consequently, I comment on the modification of the manuscript after "minor revisions" according to my previous recommendations.
#1 The authors should modify the conclusion of the abstract and avoid statements like "We believe”.
I recommend drawing wording from the summary at the end of the manuscript
The authors have edited the abstract as recommended, including the removal of inappropriate wording.
#2 Authors should add to the Materials and methods section, data on the number of technical and
biological replicates of relevant experiments from which they draw conclusions.
The authors have modified part „Materials and methods“ and also added the information I requested concerning the number of repetitions of the measurements, hence the reliability of the results.
#3 The authors should add whether the findings are applicable to human mammary adenocarcinoma
and provide references for their conclusions.
The authors added the applicability of the results towards human cells, including the reference
#4 For be􀄴er orientation in TEM fig. 4 I would divide the subfigures into more parts (more fig.) so
that the legend is closer to the figures themselves, which will improve orientation. Alternatively, the
authors may choose another way to improve the orientation of these figures.
The authors have improved the readability of the figures.
Author Response
Comment: The authors have incorporated my comments into the submitted manuscript. Consequently, I comment on the modification of the manuscript after "minor revisions" according to my previous recommendations.
#1 The authors should modify the conclusion of the abstract and avoid statements like "We believe”.
I recommend drawing wording from the summary at the end of the manuscript
The authors have edited the abstract as recommended, including the removal of inappropriate wording.
#2 Authors should add to the Materials and methods section, data on the number of technical and
biological replicates of relevant experiments from which they draw conclusions.
The authors have modified part „Materials and methods“ and also added the information I requested concerning the number of repetitions of the measurements, hence the reliability of the results.
#3 The authors should add whether the findings are applicable to human mammary adenocarcinoma
and provide references for their conclusions.
The authors added the applicability of the results towards human cells, including the reference
#4 For be?er orientation in TEM fig. 4 I would divide the subfigures into more parts (more fig.) so
that the legend is closer to the figures themselves, which will improve orientation. Alternatively, the
authors may choose another way to improve the orientation of these figures.
The authors have improved the readability of the figures.
Response: We are glad that the reviewer liked our responses and adjustments to the manuscript. Thank you for your time to revise our draft and help us improve the manuscript.
Reviewer 2 Report (Previous Reviewer 3)
Comments and Suggestions for Authors
Dear authors,
Thank you for your consideration of the comments. I would like to recommend accepting the manuscript following the revisions.
Sincerely,
Author Response
Comment: Thank you for your consideration of the comments. I would like to recommend accepting the manuscript following the revisions.
Response: We are glad that the reviewer liked our responses and adjustments to the manuscript. Thank you for your time to revise our draft and help us improve the manuscript.
Reviewer 3 Report (New Reviewer)
Comments and Suggestions for Authors
This manuscript demonstrates the trace/pathway of BPAuNPs under the 4T1 breast cancer cell line. Authors use TEM to visualize nanoparticle internalization and use qPRC to assess functional delivery by monitoring the suppression of miR-135b. The research is interesting, and I would recommend that this manuscript be published if the author could address my concerns below.
- The numbering of each section (multilevel numbering) needs to be clarified and corrected.
- A control experiment based on naked aitimiR-135b is suggested to understand that AuNPs are necessary for delivery.
- The manuscript lacks a quantitative uptake control. The uptake was only evaluated via TEM on a small number of cells, which raises questions about the statistical confidence.
Overall, this manuscript is in a scientific shape, and I would recommend major/minor revision before the author can elaborate on my concerns.
Author Response
Comment 1: The numbering of each section (multilevel numbering) needs to be clarified and corrected.
Response 1: Thank you for pointing this out. The sections and individual headings are now correctly numbered.
Comment 2: A control experiment based on naked antimiR-135b is suggested to understand that AuNPs are necessary for delivery.
Response 2: The antimiRs have been used in our laboratory in order to inhibit the target miR-21 or miR-135b mostly in murine breast or colon cancer modes. The breast cancer model has been derived from the 4T1 cell line used in this study. We experienced that the naked antimiR cannot effectively pass into a cytoplasm of specialized cells such as 4T1 without a carrier (commercial transfection reagent, nanodiamonds, or gold nanoparticles). On the other hand, uptake of naked synthetic short RNAs has been reported under specific conditions. Mainly dendritic cells / phagocytizing cells can uptake naked RNAs via macropinocytosis due to their nature. We are also aware that there are studies regarding spontaneous uptake (gymnosis) of so-called locked RNAs without the need of being carried by a carrier or vesicle. However, as summarized in a recent review (*), the efficacy of gymnosis is low and the internalized RNAs remained functional only if the samples were also treated with an RNase inhibitor. Therefore, we did not involve naked RNA treatments in the current report.
* Castellano M. et al. Cell Genom. 2025 May 14;5(5):100874. doi: 10.1016/j.xgen.2025.100874.
Comment 3: The manuscript lacks a quantitative uptake control. The uptake was only evaluated via TEM on a small number of cells, which raises questions about the statistical confidence.
Response 3: Thank you for pointing out the quantitation. We focused primarily on a qualitative analysis but we did involve quantitation based on TEM as requested by the previous reviewer (sections 2.2.2. and 4.4.). We are aware that such analysis suffers from very low sample numbers. In parallel, we performed qPCR though that serves as a quantitative method enabling us to evaluate the amount of internalized AuNPs. This statement is based on optimized ratio and time for incubation of the AuNPs with short RNAs (developed in our laboratory within previous research studies over the past 5 years and ensuring similar antimiR load on each NP). We added following text (new sections 2.3.2. and 4.5.) in the manuscript that details this analytical approach and the results are summarized in Table 4.
Section 4.5.: "Since we use optimized ratio and incubation time for loading antimiR onto AuNPs, we expect similar load of the antimiR per each AuNP. Successful delivery of the antimiR is demonstrated by the effective but not complete knock-down of the target miR 135b once the cells were treated with the AuNPs-antimiR 135b only. Using the non-treated cells as reference sample we can quantify the efficacy of miR 135b knock-down reflecting also the amount of internalized AuNPs in cell cytoplasm."
Section 2.3.2.: "Quantitation of the AuNPs uptake with respect to the cargo efficacy. If we consider that each AuNP can carry similar amount of the antimiR, then decreased level of target miR 135b roughly reflects the number of conjugates in cytoplasm. Efficacy of the miR 135b knock-down was calculated as ratios of normalized miR 135b levels in samples and non-treated cells. The best efficacy in miR 135b knock-down was achieved using the AuNPs-antimiR 135b conjugate, followed by the AuNPs-antimiR 135b-Tf and X-tremeGENE™ HP DNA Transfection Reagent. Interestingly, the chlorpromazine treatment resulted in lower efficacy of the AuNPs-antimiR 135b conjugate (83% versus 49%) but its effect on the AuNPs-antimiR 135b-Tf had slightly adverse trend 54% versus 64%. Increased efficacy in the AuNPs-antimiR 135b was also detected after amiloride treatment (18%). However, the effects of chlorpromazine and amiloride on the AuNPs-antimiR 135b are borderline due to higher SDs.
Table 4. Knock-down efficacy of the target miR 135b with respect to non-treated cells. Values are presented as a percentage of target miR 135b level normalized to its reference level in non-treated cells. HP: X-tremeGENE™ HP DNA Transfection Reagent, Ami: amiloride hydrochloride, Ch: chlorpromazine, and Nys: nystatin. Remarkable differences are in bold.
A comparison of the miR 135b knock-down among non-treated cells and those differently inhibited internalization pathways showed that cells pretreated with amiloride hydrochloride or nystatin exhibited similar miR 135b level as control non-treated cells. Opposing, the cells pretreated with chlorpromazine showed effective miR 135b inhibition which means that the AuNPs-antimiR 135b effectively entered the cells via alternative, non-blocked pathways. Thus, clathrin-independent pathways such as caveolin-mediated endocytosis and macropinocytosis are the primary methods of the AuNPs internalization delivering functional cargo into cell cytoplasm. "
Reviewer 4 Report (New Reviewer)
Comments and Suggestions for Authors
In the present manuscript, the authors report the results of their investigation aimed at understanding the cellular trafficking of biologically produced gold nanoparticles (BPAuNP) in 4T1 cells. They employed various experimental techniques to gather relevant information, although the analysis is primarily qualitative. The topic of the manuscript is certainly interesting, and the findings could provide valuable insights into the behaviour of BPAuNP.
Given my field of expertise, I focused my comments on the experimental aspects of the manuscript. Since the authors utilised multiple experimental techniques, it is essential to describe them accurately, along with the conditions under which they were employed. A brief paragraph in the Materials and Methods section listing the experimental apparatus and detailing its technical features and applications in this investigation would be beneficial.
Below is a brief list of points for the authors to address:
- In paragraph 1.1.1, the spectrophotometry measurements are described. Please include the model of the Tecan instrument used and details of the spectrophotometer and measurement procedures. For example, the spectral resolution of the instrument is relevant since, on page 8, small changes in peak position are cited to support claims regarding the conjugation of AuNPs to different molecules.
- On line 24 of page 3, zeta potential measurements are mentioned, but no information about the experimental apparatus is provided.
- Similarly, on line 27 of page 3, the same issue arises for the Transmission Electron Microscopy (TEM) measurements. While some details about the apparatus are provided on line 22 of page 6, it’s unclear whether the authors are using the same equipment.
- In Figure 1, the photographs labelled A, B, C, and D do not convey significant information. Please consider improving these images for clarity and impact.
Author Response
Comment 1: In paragraph 1.1.1, the spectrophotometry measurements are described. Please include the model of the Tecan instrument used and details of the spectrophotometer and measurement procedures. For example, the spectral resolution of the instrument is relevant since, on page 8, small changes in peak position are cited to support claims regarding the conjugation of AuNPs to different molecules.
Response 1: Thank you for pointing this out. We added the instrument details “Infinite 200 PRO UV-visible spectrophotometer (Tecan, Männedorf, Switzerland)” into the text where it appears for the first time (section 4.2.1.) and then we refer to it as Tecan spectrophotometer.
Comment 2: On line 24 of page 3, zeta potential measurements are mentioned, but no information about the experimental apparatus is provided.
Response 2: Thank you for pointing this out. We added following text (section 4.2.3. and 4.1.) to the manuscript.
section 4.2.3.: “To characterize size and charge of the AuNP conjugates, triplicates of test samples (AuNPs-antimiR 135b and AuNPs-antimiR 135b-Tf), along with control triplicates (AuNPs), were assessed for their size and zeta potential using DLS and ELS with the Zetasizer as detailed in 4.1.”
section 4.1.: "To characterize size and charge of the AuNPs, we employed dynamic light scattering (DLS) through a non-invasive backscattering method, and electrophoretic light scattering (ELS), respectively, with a Zetasizer Ultra instrument (Malvern Panalytical, Malvern, UK). The measurements employed a refractive index of 0.18, an absorbance of 3.43, and utilized double-distilled water (ddHâ‚‚O) as the dispersant. For size distribution assessment, an ultra-low-volume quartz cuvette (ZEN2112) was used, with data collected in a backscatter mode. Zeta potential measurements were performed using a DTS1070 zeta cell with folded capillaries, all at a controlled temperature of 25 °C [30]. Samples were measured in triplicates."
Comment 3: Similarly, on line 27 of page 3, the same issue arises for the Transmission Electron Microscopy (TEM) measurements. While some details about the apparatus are provided on line 22 of page 6, it’s unclear whether the authors are using the same equipment.
Response 3: Yes, we used the same equipment. We restructured and unified the text throughout the manuscript so it lines with the journal standards. This comment is addressed now in a section 4.4.
section 4.4.: "4T1 cells (2 × 10 3 cells/mL) were cultured in the same working medium as mentioned above, but on round cover slips (12 mm) in a 24-well plate. After 48 hours of incubation, the cells reached 60-70 % confluence in 500 µL of fresh culture medium. The wells containing cover slips were then divided into 7 groups (each group in triplicates). Two groups were treated with the above-mentioned amounts of nystatin, or amiloride hydrochloride, and two groups after treatment with chlorpromazine. After 30 minutes of incubation, the chemicals were washed twice with PBS, and the wells were refilled with 500 µL of the working medium. In parallel, 3 groups (each in triplicate) remained untreated and served as controls. After pretreatment with chemicals, two groups (nystatin and amiloride hydrochloride) were incubated with 2.5 µL of AuNPs-antimiR 135b. The third and fourth groups (chlorpromazine treatment) were incubated either with AuNP-antimiR 135b or AuNP-antimiR 135b-Tf. The control groups included cells without any treatment, and two groups incubated with 2.5 µL of AuNP-antimiR 135b or AuNP-antimiR 135b-Tf. The incubation time with the AuNPs was 2 hours, then cells were washed with PBS, and fixed with 1% glutaraldehyde (GA) containing 2.5% Paraformaldehyde/Sodium cacodylate buffer (PFA/SB buffer). After 1 hour of fixation at 4°C and three washings with SB buffer, 1% OsO4 in SB was added, followed by another hour of incubation in the dark at room temperature. The plate was then washed with SB buffer, rinsed with water three times, and incubated with various concentrations of acetone in water (30 %, 50 %, 70 %, 90 %, and 95 %) before finally using water-free acetone. The samples were embedded and polymerized using Epon-Durcupan. The resin blocks were cut into 85 nm slices using an ultramicrotome (Leica EM UC6, Prague, Czech Republic) and placed on copper grids (square, 200 mesh, Agar Scientific, Essex, UK). Images were acquired using a transmission electron microscope Jeol JEM 1400 Flash (120 kV), operated at 80 kV, and equipped with a tungsten cathode and bottom-mounted FLASH 2kx2k CMOS camera. The same AuNPs used for cell experiments were also analyzed alone using TEM. For this purpose, 2 mL of the sample was placed on a glow-discharge-activated grid (30 seconds, 1 kV, 10 mA). After air drying, the samples were analyzed by TEM (JEOL JEM-F200), operated at 200 kV and equipped with a Cold FEG and TVIPS XF416 camera. To prove the presence of elemental Au in the samples, the control cells incubated with AuNPs were checked using a JED 2300 X-ray spectrometer to obtain EDS spectra from the zones containing AuNPs.
The spectra were then analyzed using Jeol AnalysisStation software. TEM photomicrographs were used to roughly evaluate the amount of internalized AuNPs per cell in each group (i.e., AuNPs-antimiR-135b and AuNPs-antimiR-135b-Tf). For this purpose, the cells (n=3 / group) were first examined at a magnification of 5 µm, and then AuNPs were counted in at least three random areas of each cell at a magnification of 500 nm. The number of AuNPs was recorded, and the average total number of nanoparticles was calculated for each group. Additionally, colocalization and the quantities of AuNPs within early endosomes and late endosomes/lysosomes were assessed. To compare the results, data were compared using analysis of variance (ANOVA) in SPSS software (available at ANOVA Calculator - One Way ANOVA and Tukey HSD test). p-values of ≤0.05 were considered as significant."
Comment 4: In Figure 1, the photographs labelled A, B, C, and D do not convey significant information. Please consider improving these images for clarity and impact.
Response 4: Thank you for your comment. We revised Figure 1 to add clarity.
Reviewer 5 Report (New Reviewer)
Comments and Suggestions for Authors
Reviewer:
Recommendation: This paper may be publishable after minor revision; I would appreciate being invited to review the revised version.
Comments:
This manuscript offers a comprehensive investigation into the cellular uptake and trafficking of anti-miR conjugated bio-produced gold nanoparticles in 4T1 cells. By integrating TEM imaging and qPCR, the study effectively elucidates both the mechanisms of internalization and the delivery of active oligonucleotide cargo. The use of specific endocytosis inhibitors, alongside clear experimental methods, provides strong mechanistic insight and highlights key differences between biologically produced and chemically synthesized nanoparticles. Emphasizing the delivery of functionally intact cargo is a notable strength and has important implications for therapeutic nano medicine. Addressing limitations such as the use of a single cell line and providing further clarification on quantitative analyses would make this strong contribution even more impactful. However, several key aspects require clarification or strengthening to better position the significance and rigor of the work. Please find below my major and minor comments.
.Regarding specific questions, I would like the authors to address the following points prior to acceptance:
Major Comments:
- The statistical power of TEM quantification is limited by the low sample size; please clarify the rationale and consider increasing replicates.
- The fate of internalized nanoparticles and cargo in lysosomes is not fully explored; co-localization studies with lysosomal markers could clarify this aspect. Please add the marker controls.
- Only short-term uptake is studied; extending experiments to include long-term stability and functional outcomes would strengthen the work.
- Off-target effects and potential cytotoxicity of chemical inhibitors warrant additional discussion and experimental confirmation.
- Some figures and tables lack clear statistical annotation and group definitions; improved clarity and explanation are needed for these data presentations.
Minor Comments:
- Abbreviations and terminology for nanoparticles should be consistent throughout the text to avoid confusion.
- All figures and tables should be referenced in sequence and clearly within the main text.
- Minor language issues and awkward phrasing are present; additional proofreading or language editing is advised.
- Methodological details such as reagent sources and washing steps should be clearly described to enhance reproducibility.
- The abstract should more explicitly state the novelty of using bio-produced nanoparticles to emphasize the manuscript’s significance.
Author Response
Comment 1: The statistical power of TEM quantification is limited by the low sample size; please clarify the rationale and consider increasing replicates.
Response 1: Thank you for pointing out the quantitation. We focused primarily on a qualitative analysis but we did involve quantitation based on TEM as requested by the previous reviewer (sections 2.2.2. and 4.4.). We are aware that such analysis suffers from very low sample numbers. In parallel, we performed qPCR though that serves as a quantitative method enabling us to evaluate the amount of internalized AuNPs. This statement is based on optimized ratio and time for incubation of the AuNPs with short RNAs (developed in our laboratory within previous research studies over the past 5 years and ensuring similar antimiR load on each NP). We added following text (new sections 2.3.2. and 4.5.) in the manuscript that details this analytical approach and the results are summarized in Table 4.
Section 4.5.: Since we use optimized ratio and incubation time for loading antimiR onto AuNPs, we expect similar load of the antimiR per each AuNP. Successful delivery of the antimiR is demonstrated by the effective but not complete knock-down of the target miR-135b once the cells were treated with the AuNPs-antimiR-135b only. Using the non-treated cells as reference sample we can quantify the efficacy of miR-135b knock-down reflecting also the amount of internalized AuNPs in cell cytoplasm.
Section 2.3.2.: "Quantitation of the AuNPs uptake with respect to the cargo efficacy. If we consider that each AuNP can carry similar amount of the antimiR, then decreased level of target miR 135b roughly reflects the number of conjugates in cytoplasm. Efficacy of the miR 135b knock-down was calculated as ratios of normalized miR 135b levels in samples and non-treated cells. The best efficacy in miR 135b knock-down was achieved using the AuNPs-antimiR 135b conjugate, followed by the AuNPs-antimiR 135b-Tf and X-tremeGENE™ HP DNA Transfection Reagent. Interestingly, the chlorpromazine treatment resulted in lower efficacy of the AuNPs-antimiR 135b conjugate (83% versus 49%) but its effect on the AuNPs-antimiR 135b-Tf had slightly adverse trend 54% versus 64%. Increased efficacy in the AuNPs-antimiR 135b was also detected after amiloride treatment (18%). However, the effects of chlorpromazine and amiloride on the AuNPs-antimiR 135b are borderline due to higher SDs.
A comparison of the miR 135b knock-down among non-treated cells and those differently inhibited internalization pathways showed that cells pretreated with amiloride hydrochloride or nystatin exhibited similar miR 135b level as control non-treated cells. Opposing, the cells pretreated with chlorpromazine showed effective miR 135b inhibition which means that the AuNPs-antimiR 135b effectively entered the cells via alternative, non-blocked pathways. Thus, clathrin-independent pathways such as caveolin-mediated endocytosis and macropinocytosis are the primary methods of the AuNPs internalization delivering functional cargo into cell cytoplasm.
Table 4. Knock-down efficacy of the target miR 135b with respect to non-treated cells. Values are presented as a percentage of target miR 135b level normalized to its reference level in non-treated cells. HP: X-tremeGENE™ HP DNA Transfection Reagent, Ami: amiloride hydrochloride, Ch: chlorpromazine, and Nys: nystatin. Remarkable differences are in bold."
Comment 2: The fate of internalized nanoparticles and cargo in lysosomes is not fully explored; co-localization studies with lysosomal markers could clarify this aspect. Please add the marker controls.
Response 2: Thank you for your suggestion. Within this manuscript we employed TEM analysis to detect intracellular location of the individual particles instead of endosome/lysosome marker staining. The mechanism of cargo release into cytoplasm and fate of loaded/unloaded nanoparticles, particularly with respect to different culture conditions or cargo properties is important information that will be assessed as complex in future. The release mechanism here will not be as straightforward as in synthetic nanoparticles with defined surface modification. It is due to the presence of a not-yet defined capping agent that is formed on the particle surface during the biological production. We are aware that the biological nanoparticles, even though they present an attractive and potent tool, are still not well-characterized and their properties rely on particular production conditions.
Comment 3: Only short-term uptake is studied; extending experiments to include long-term stability and functional outcomes would strengthen the work.
Response 3: Thank you for this comment. Yes, the long-term stability can be a limiting factor for subsequent carrier applications and readers may be interested in that parameter. From our experience, the AuNPs remain stable at 4°C over one year which means that it is feasible to perform one study from the same batch of nanoparticles. We added this information including spectra of long term (over 12 months) stored AuNPs and their conjugate with antimir and Tf. It can be found as Supplementary figure 1.
Reference in the main text is in section 2.1.1.: „Produced AuNPs and AuNPs-antimiR 135b-Tf remained stable during the whole duration of the study and the absorbance spectra of long-term (over 12 months) stored samples can be seen in Supplementary Figure 1.“
Comment 4: Off-target effects and potential cytotoxicity of chemical inhibitors warrant additional discussion and experimental confirmation.
Response 4: Yes, off-targeting is an important issue. To minimize it, we used optimized concentrations of individual components. We added following text into discussion:
“Using chemical inhibitors to study different internalization pathways may trigger off-target effects and potential cytotoxicity of the chemical inhibitors. In the current study, we employed three different chemical treatments: nystatin at a final concentration of 50 µM, amiloride hydrochloride hydrate at 0.5 mM, and chlorpromazine hydrochloride at 0.03 µM. The concentrations used for each chemical were below their respective cytotoxic levels [17, 18]. Two of the chemicals mentioned above (nystatin and amiloride hydrochloride) exhibit known off-target activities. For instance, nystatin is primarily used as an antifungal agent but it also exhibits off-target effects such as binding to cholesterol, disrupting cholesterol-enriched membrane microdomains (lipid rafts), and altering internalization pathways [27]. We have leveraged this off-target activity of nystatin for our current research. Other off-target effects of nystatin, including nephrotoxicity, pro-inflammatory responses, and impacts on axon growth and regeneration, are observed mainly in vivo and are not relevant to our in vitro study. Amiloride hydrochloride hydrate is a pyridine compound used to treat hypertension and congestive heart failure. It functions as a potassium-sparing diuretic by blocking epithelial sodium channels (ENaC) in the distal nephron of the kidneys. Additionally, amiloride's inhibition of Na+/H+ exchange may indirectly influence macropinocytosis by altering sub membranous pH, which is essential for the activity of GTPases involved in actin remodeling—a critical step in macropinosome formation [28]. While its primary mechanism of action as a drug is modulation of renal ion channels, amiloride can also inhibit macropinocytosis. This effect is likely due to changes in sub membranous pH resulting from its inhibition of Na+/H+ exchange, representing an off-target action that we leveraged in our study.
Another chemical we used in our study was chlorpromazine hydrochloride, an antipsychotic medication primarily used to treat psychotic disorders such as schizophrenia. As a cationic amphiphilic drug, chlorpromazine exerts its effects by binding to clathrin and AP2, thereby inhibiting their proper assembly into clathrin-coated vesicles at the plasma membrane. Additionally, it may impede dynamin, a GTPase essential for clathrin-mediated endocytosis. Chlorpromazine has also been shown to disrupt basolateral actin cytoskeleton dynamics, which are important for AQP2 endocytosis in kidney epithelial cells. However, detailed studies on the off-target effects of this drug are limited [29].”
Comment 5: Some figures and tables lack clear statistical annotation and group definitions; improved clarity and explanation are needed for these data presentations.
Response 5: Thank you for your suggestion. We revised the legends and description in the Tables and Figures to add clarity.
Comment 6: Abbreviations and terminology for nanoparticles should be consistent throughout the text to avoid confusion.
Response 6: Thank you for pointing this out. We unified the terminology and abbreviation throughout the manuscript.
Comment 7: All figures and tables should be referenced in sequence and clearly within the main text.
Response 7: Thank you for pointing this out. We checked and corrected placement of the Figures and Tables throughout the manuscript.
Comment 8: Minor language issues and awkward phrasing are present; additional proofreading or language editing is advised.
Response 8: Thank you for your comment. We employed English editing by a native English speaker to improve readability of the text.
Comment 9: Methodological details such as reagent sources and washing steps should be clearly described to enhance reproducibility.
Response 9: Thank you for pointing this out. We have added the details in the materials and method part - Section 4.
Comment 10: The abstract should more explicitly state the novelty of using bio-produced nanoparticles to emphasize the manuscript’s significance.
Response 10: Thank you for your suggestion. We revised and restructured the abstract along with the journal guidelines and added suggested parts.
"Background/Objectives: Bio-produced gold nanoparticles (AuNPs) are effective carriers of short RNAs into specialized mammalian cells. Their potential application is still limited by scarce knowledge on their uptake and intracellular fate. Gold nanoparticles that are not biologically produced (NB-AuNPs) enter specialized cells primarily via clathrin-dependent endocytosis. Unlike the NB-AuNPs, the bio AuNPs possess natural surface coatings that significantly alter the AuNPs properties. Our research aimed to reveal the cellular uptake of the AuNPs with respect to delivering a functional RNA cargo. Methods: The AuNPs were conjugated with short inhibitory RNA specific to miR 135b. Mammary cancer cells 4T1 were pretreated with inhibitors of caveolin- and clathrin- mediated endocytosis and macropinocytosis. AuNPs uptake, fate, and miR 135b knock-down were assessed with TEM and qPCR. Results: The AuNPs-antimiR 135b conjugates entered 4T1 cells via all the tested pathways and could be seen inside the cells in early and late endosomes as well as cytoplasm. In contrast to the clathrin-dependent pathway, the caveolae-mediated endocytosis and the macropinocytosis of the AuNPs resulted in effective targeting and reduction of the miR 135b. Conclusion: The bio-produced AuNPs can effectively enter mammalian cells simultaneously by different endocytic pathways but delivery of functional cargo is not achieved via the clathrin-dependent endocytosis."
Round 2
Reviewer 4 Report (New Reviewer)
Comments and Suggestions for Authors
The authors satisfactorily revised their manuscript by taking into account my comments.
This manuscript is a resubmission of an earlier submission. The following is a list of the peer review reports and author responses from that submission.
Round 1
Reviewer 1 Report
Comments and Suggestions for Authors
The authors of the article “ Qualitative cellular tracking of antimiR conjugated bio-produced gold nanoparticles” give an explanation of the process of cellular internalization of gold nanoparticles (BPAuNPs) and their further processing at the cellular level. Understanding this process is important in terms of preparing safe anticancer drugs based on BPAuNPs conjugated with active anti-tumor cargo.
Minor revisions:
#1 The authors should modify the conclusion of the abstract and avoid statements like "We believe”. I recommend drawing wording from the summary at the end of the manuscript
#2 Authors should add to the Materials and methods section, data on the number of technical and biological replicates of relevant experiments from which they draw conclusions.
#3 The authors should add whether the findings are applicable to human mammary adenocarcinoma and provide references for their conclusions.
#4 For better orientation in TEM fig. 4 I would divide the subfigures into more parts (more fig.) so that the legend is closer to the figures themselves, which will improve orientation. Alternatively, the authors may choose another way to improve the orientation of these figures.
Reviewer 2 Report
Comments and Suggestions for Authors
Comments on ncrna-3647235:
The authors report on the cellular uptake mechanisms of biologically produced gold nanoparticles (BPAuNPs) conjugated with antimiR-135b into 4T1 mouse breast cancer cells. The study tries to show that BPAuNPs can deliver intact cargo inside cells by different endocytosis pathways and retain antimiR activity using TEM and qPCR techniques. A quick search in databases shows that the presented work is not novel and the novelty is incremental. Studies on endocytosis pathways of gold NPs are well-documented including RNA cargo and using inhibitors. Several previously published papers, which some of them are referred by the authors too (e.g., references 21, 25, 26) have also examined endocytosis routes using similar methods, although for chemically synthesized gold NPs. Therefore, I don’t recommend publishing this manuscript in the present form due to lack of novelty and other major issues present in the manuscript which some of them are listed below for authors information:
- The authors need to clearly discuss what differentiates their findings from existing reports. For example: Does the biogenic capping layer affect the endosomal escape or RNA protection? Is the uptake route altered because of fungal-sourced biocoating? And many other similar questions.
- The TEM analysis is qualitative only. No quantification of NPs number per cell, or endosome/lysosome colocalization (e.g., via confocal) is presented.
- No deep characterizations of the produced NPs is performed, as authors re not using commercially available NPs.
- No tracking of endosomal escape, which is critical to determine whether qPCR suppression comes from cytoplasmic release or endosomal degradation is presented. In fact, the major issue is that qPCR alone can not prove cytoplasmic release.
- Authors discuss that the decrease in miR-135b is related to antimiR activity, however there is no control showing stability of the antimiR in lysosomal vs. cytoplasmic conditions.
- Also, no mechanistic insights into lysosomal degradation are presented.
- Although figures 4 and 5 contain some details but there is no quantification (e.g., how many cells had uptake in each group) and everything is qualitative.
Comments on the Quality of English Language
Mentioned above
Reviewer 3 Report
Comments and Suggestions for Authors
Dear authors,
This study is well organized, and the manuscript is well composed. The findings have the potential to engage readers. However, there are still minor issues that require attention.
- The first sentence of the abstract fails to adequately inform readers regarding the content of the paper. A concise introduction should be incorporated into the abstract.
- The focus on the synthesis and characterization of AuNPs should be sharpened.
Although the AuNPs were synthesized following a prior publication, a brief description of the techniques and materials utilized in the synthesis is still necessary. What methods were employed for the characterization of AuNPs?
- A TEM image of the nanoparticles should be included, along with a graph depicting the size distribution.
- The TEM images of the treatment on 4T1 cells appeared to be quite impressive. Is it possible to perform any quantification to illustrate the differences between the conditions without and with AuNPs?
Sincerely,